# Quantitative Bio-Mapping of *Salmonella* and Indicator Organisms at Different Stages in a Commercial Pork Processing Facility

**DOI:** 10.3390/foods11172580

**Published:** 2022-08-25

**Authors:** Rossy Bueno López, David A. Vargas, Reagan L. Jimenez, Diego E. Casas, Markus F. Miller, Mindy M. Brashears, Marcos X. Sanchez-Plata

**Affiliations:** International Center for Food Industry Excellence, Department of Animal and Food Sciences, Texas Tech University, Lubbock, TX 79409, USA

**Keywords:** *Salmonella* enumeration, pork bio-mapping, statistical process control, new swine inspection system

## Abstract

The purpose of this study was to develop a quantitative baseline of indicator organisms and *Salmonella* by bio-mapping throughout the processing chain from harvest to final product stages within a commercial conventional design pork processing establishment. Swab samples were taken on the harvest floor at different processing steps, gambrel table, after polisher, before final rinse, after the final rinse, post snap chill, and after peroxyacetic acid (PAA) application, while 2-pound product samples were collected for trim and ground samples. The samples were subjected to analysis for indicator microorganism enumeration, Aerobic Count (AC), Enterobacteriaceae (EB), and generic *Escherichia coli* (EC), with the BioMérieux TEMPO^®^. *Salmonella* prevalence and enumeration was evaluated using the BAX^®^ System Real-Time *Salmonella* and the SalQuant™ methodology. Microbial counts were converted to Log Colony-forming units (CFU) on a per mL, per g or per sample basis, presented as LogCFU/mL, LogCFU/g and LogCFU/sample, prior to statistical analysis. All indicator microorganisms were significantly reduced at the harvest floor (*p*-value < 0.001), from gambrel table to after PAA cabinet location. The reduction at harvest was 2.27, 2.46 and 2.24 LogCFU/mL for AC, EB and EC, respectively. Trim sample values fluctuated based on cut, with the highest average AC count found at neck trim (2.83 LogCFU/g). Further process samples showed the highest AC count in sausage with a mean of 5.28 LogCFU/g. EB counts in sausage (3.19 LogCFU/g) showed an evident increase, compared to the reduction observed at the end of harvest and throughout trim processing. EC counts showed a similar trend to EB counts with the highest value found in sausage links (1.60 LogCFU/g). Statistical microbial process control (SPC) parameters were also developed for each of the indicator microorganisms, using the overall mean count ($\overset{=}{X}$), the Lower control limit (LCL) and Upper control limit (UCL) at each sampling location. For *Salmonella* prevalence, a total of 125/650 samples were found positive (19%). From those positive samples, 47 samples (38%) were suitable for enumeration using the BAX^®^ System SalQuant™, the majority detected at the gambrel table location. From those enumerable samples, 60% were estimated to be between 0.97 and 1.97 LogCFU/sample, while the rest (40%) were higher within the 2.00–4.02 LogCFU/sample range. This study provides evidence for the application of indicator and pathogen quantification methodologies for food safety management in commercial pork processing operations.

## 1. Introduction

Pork ranks second as the most consumed meat in the world with an estimate of 106.3 million tons consumed as of 2020, recently surpassed by poultry meat (131.2 million tons) [1]. For that same year, the United States’ total pork production reached 28.3 billion pounds, with a per capita consumption of pork at 50.6 pounds [2,3]. The United States has been excelling as one of the world’s main exporters of pork meat. According to the National Pork Producers Council, exports of pork and pork related products yield more than 2.2 million metric tons annually [4]. Economically, this translates to $7.7 billion for the value of U.S. pork and pork product exports, showing an 11% increase when compared to the previous year [5]. Based on numbers from the United States Department of Agriculture (USDA) in 2019, 129.9 million farmed pigs were slaughtered for food in the U.S. Furthermore, the pork industry’s market size in the US is estimated at $19 billion [6]. Pork consumption in the US occurs mainly in the form of fresh pork cuts (e.g., chops, roasts, steaks, ribs, fresh ham) and the remaining as processed pork (comminuted pork, such as sausage), hot dogs, bacon, and ready-to-eat products (e.g., cooked ham, lunch meats).

It has been estimated that every year in the United States 48 million people become ill because of foodborne illnesses. In 20% of these cases (9.4 million illnesses) a specific pathogen can be attributed as the cause [7]. The incidence of domestically acquired foodborne illnesses caused by non-typhoidal *Salmonella* in the US each year is reflected with 1,027,561 cases (11%), 19,336 hospitalizations (35%), and 378 deaths (28%) [7]. In terms of economic burden, *Salmonella* ranks first among 15 pathogens included in a USDA report with a $3.7 billion economic burden per year [8]. According to the foodborne illness attribution estimates for 2019, published by the Interagency Food Safety Analytics Collaboration (IFSAC) in October 2021 with efforts from the Centers for Disease Control and Prevention (CDC), the U.S. Food and Drug Administration (FDA), and the USDA’s Food Safety Inspection Service (FSIS), 75.9% of illnesses were attributed to seven food categories: chicken, fruits, pork, seeded vegetables, other produce, turkey, and eggs. Furthermore, 12.8% of that breakdown is attributed to pork [9]. This makes pork the second highest contributor of Salmonellosis in FSIS regulated products. The report includes encompassing data from 1998 to 2019 which further presents an ample view on outbreak-based attribution estimates on the percentage of illnesses caused by four pathogens: *Salmonella*, *Escherichia coli* O157, *Listeria monocytogenes*, and *Campylobacter*.

The late 1990s was marked by the introduction of the Hazard Analysis Critical Control Point (HACCP) regulations (9 CFR Part 417) for all USDA inspected meat and poultry facilities [10]. The regulation required testing of carcasses as the basis of performance standards [10]. Data evaluation led to the conclusion that carcasses may not be the best sampling point to assess the *Salmonella* status of consumer products [11] As a result, FSIS switched to sampling beef trim and pork products as part of the process of modernized systems that support HACCP-based principles. The goal of modernized inspection is driven by FSIS science-based and data-driven efforts, encompassing inspection sampling and verification data points with risk assessments [12]. The role of FSIS inspectors is to verify an establishment’s food safety system is working to ensure the safety of products by conducting inspection and sampling. In addition, establishments are required to test for indicators as part of their food safety management system to help facilitate FSIS inspectors’ verification of controls by the food industry to foresee and prevent foodborne hazards, especially pathogens. Each carcass is also evaluated for visible contamination and adulteration. With modernized inspection systems, *Salmonella* reduction is targeted with noncompliance rates used to determine thresholds and by prioritizing establishments for public health risk evaluations (PHRE) [12].

*Salmonella* performance standards are set to hold the industry responsible for pathogen control by implementing best processing practices, HACCP implementation and validating intervention technologies to ensure products are safe for the consumer [12]. On February 2022, FSIS proposed performance standards for *Salmonella* in raw comminuted pork and pork cuts [13]. The prevalence of *Salmonella* in raw comminuted pork and pork cuts is estimated at 30% and 9%, respectively [13]. These thresholds are used to categorize operations on the basis of performance. In April 2015, FSIS introduced the Raw Pork Products Exploratory Sampling Program (RPPESP) with the intent to collect data on the presence of *Salmonella*, *Campylobacter* spp., *Toxoplasma gondii*, *Yersinia enterocolitica*, Methicillin-Resistant *Staphylococcus aureus*, Shiga toxin-producing *E. coli*, and indicator organisms in pork products [14]. The four indicators tested were Aerobic Count (AC), Enterobacteriaceae (EB), coliforms, and *Escherichia coli* (EC). The project involved retail sampling, the first phase exploratory sampling at slaughter and processing establishments, samples taken during the transition phase, and a second phase of exploratory sampling. The eligible product categories for sampling were intact cuts (bone-in and boneless), non-intact cuts (bone-in and boneless), and comminuted products (ground pork, sausage, patties, and other formed products, mechanically separated, Advanced Meat Recovery or AMR, and other comminuted pork) [14]. Data from these nationwide sampling studies have been used as the basis for standards to reduce the level of *Salmonella* in specific raw pork products.

The HACCP-based Inspection Model Project (HIMP) led to the foundational basis for the New Swine Inspection System (NSIS) pilot in 2014 [15,16]. The HIMP system differs from the traditional inspection system by providing the establishment with more control for food safety and activities linked to consumer protection while agency personnel are involved in carcass and verification system activities [15]. HIMP provides FSIS and other stakeholders with more than 15 years of data that led to the final rule “Modernization of Swine Slaughter Inspection (84 FR 52300) [17]. This final rule established an optional new inspection system for market hog slaughter establishments (NSIS) as of December 2019 [18]. The HIMP participating plants in the market hog category involved five plants [15]. Line speeds in HIMP participating plants are adjusted on a plant-to-plant basis to optimize efficiency with worker safety, animal welfare, food safety, and food quality considerations. Under the traditional inspection system, the maximum allowable line speed is 1106 head per hour (hph). HIMP plants’ line speeds varied from 885 to 1295 hogs per hour, with an average of 1099 hph. Under NSIS, plants are allowed to operate at speeds that maintain food safety, worker safety, animal welfare, and food quality. Moreover, FSIS is actively involved with in-plant verification of food safety process controls. In addition, after plant sorting activities, FSIS’s role is to inspect every animal before slaughter and every carcass after harvest [17,19].

Despite the need for pork processing establishments to prioritize a plan of action to conduct additional microbiological testing as part of NSIS and comply with the proposed USDA-FSIS *Salmonella* performance standards, there is minimal information on the implementation of indicator enumeration and *Salmonella* quantitative levels throughout the pork processing line. This is the first bio-mapping and 10-week longitudinal study that includes enumeration of *Salmonella* and indicator organisms in harvest, trim, and further process stages in a conventional design pork processing facility, operating under HIMP. The purpose of this study was to develop a baseline based on quantification of indicator organisms and *Salmonella* by bio-mapping through the processing chain from harvest to final product to demonstrate microbial control in pork processing operations implementing the New Swine Inspection System (NSIS) to establish statistical process control (SPC) parameters for food safety management.

## 2. Materials and Methods

### 2.1. Sample Collection

The study was conducted in a commercial processing facility located in the United States with a slaughter capacity of 10,400 head per day. At the time of the study, the plant was operating under NSIS. Five individual samples per location, including harvest swabs, pork trim, and ground pork, were collected over a 10-week period, involving 13 sampling locations (Harvest: gambrel table, after polisher, before final rinse, after final rinse, post snap chill, and after peroxyacetic acid (PAA) cabinet; Trim: boneless picnic, belly trim, neck trim, and loin trim; Further Process: Advanced Meat Recovery (AMR), ground brick trim, and sausage links), (*n* = 50; N = 650 samples). The pork processing line and corresponding sampling locations are presented in Figure 1. The PAA concentration at the cabinet location was 200 ppm. EZ-Reach™ Polyurethane Sponge Samplers with 10 mL HiCap Neutralizing Broth (World Bioproducts, Mundelein, IL, USA) were used for samples on the harvest floor. The carcass sample site consisted of the ham, belly, and jowl as these are the regions with the greatest chance of contamination during the slaughter/dressing procedure [20,21]. The three sites were swabbed using a single sponge per carcass. For trim and ground samples, sample collection was conducted following the protocols for Whole Pork Cuts (Intact and Non-Intact) and Comminuted Pork Aseptic Grab Sample Not in Final Packaging, respectively, per FSIS Directive Number 65–20 from the Raw Pork Products Sampling Program [22]. This consisted of using fresh, not frozen, raw pork, using a single Whirl-Pak bag (MilliporeSigma, Burlington, MA, USA) to aseptically collect two pounds of the corresponding cut or comminuted pork to fill the bag leaving 2 to 3 inches of space at the top. For AMR, finely textured pork, samples, bags were filled using a sanitized scoop and spatula. Samples were immediately chilled and shipped overnight to the International Center for Food Industry Excellence (ICFIE) Food Microbiology laboratory at Texas Tech University for microbiological analysis.

### 2.2. Sample Processing 

Swabs were processed by adding 50 mL buffered peptone water (Hygiena™, Camarillo, CA, USA) and homogenized in a stomacher (Model 400 circulator, Seward, West Sussex, UK) at 230 rpm for 30 s. For pork trim, 50 g of the specific cut were weighed into a 55 oz filtered Whirl-Pak bag and 200 mL BAX MP media (Hygiena™, Camarillo, CA, USA) was added. Trim samples were homogenized using a stomacher (Model 400 circulator, Seward, West Sussex, UK) at 230 rpm for 30 s. For ground pork samples, 50 g were weighed using a sterile scoopula into a 55 oz filtered Whirl-Pak bag and 200 mL BAX MP media was added. Ground pork samples were homogenized using a stomacher at 230 rpm for 1 min. From the primary bag, 30 mL of homogenate was aseptically transferred into a 24 oz filtered Whirl-Pak bag using a 50 mL disposable serological pipette (Fisher Scientific, Waltham, MA, USA). Additionally, 30 mL BAX MP media, containing 1 mL Quant solution (Hygiena™, Camarillo, CA, USA) was added to the 30 mL pure sample. The bag was homogenized by hand for 10 s after media addition. An aliquot of the three sample types (7 mL for swabs and 10 mL for trim and ground samples) was pulled and transferred using a 10 mL disposable serological pipette into sterile tubes for microbial indicators enumeration, prior to sample incubation for *Salmonella* enumeration and prevalence.

### 2.3. Microbial Indicators Enumeration

The TEMPO^®^ system (BioMérieux, Paris, France) was used for the enumeration of indicator microorganisms. For AC, the Association of Official Agricultural Chemists (AOAC) 121204 method was used, where TEMPO cards were incubated for 22–28 h at 35 ± 1 °C. For EC enumeration, AOAC 080603 method was followed, and cards were incubated for 22–27 h at 35 ± 1 °C. For EB enumeration, AOAC 050801 method was followed, and cards were incubated for 22–27 h at 35 ± 1 °C.

### 2.4. Salmonella Enumeration and Prevalence

Swabs and pork trim samples were immediately incubated at 42 °C for 6 h for quantification purposes. Ground pork samples were incubated at 42 °C for a 7-h period for enumeration. After incubation, the AOAC 081201 protocol for enumeration of *Salmonella* using the BAX^®^ System SalQuant™ (Hygiena, Camarillo, CA, USA) was followed. Additionally, the SalQuant™ protocols for pork trim and ground pork are part of the AOAC validation Level 2 modification to the BAX^®^ System Real-Time PCR Assay for *Salmonella* and BAX^®^ System SalQuant™ (Certification No. 081201). After obtaining a sample for enumeration protocol, samples were placed back to continue incubation at 42 °C for a period of 18–24 h (prevalence testing). If samples were not positive for BAX^®^ System SalQuant™, the BAX^®^ System RT-*Salmonella* Assay protocol for detection was followed. The BAX^®^ System Real-Time PCR Assay for *Salmonella* can be subdivided into 3 stages involving preparation, lysis, and PCR. The first stage consisted of prepping the lysis reagent and pre-heating thermal blocks to 37 °C and 95 °C. The lysis step involved the 5 µL sample transfer to cluster tubes, a heating step to 37 °C for 20 min, a subsequent heating step to 95 °C for 10 min, and cooling for 5 min. The PCR stage entailed hydrating PCR tables with 30 µL lysate and running the BAX^®^ Q7 thermocycler.

### 2.5. Statistical Analysis

All data were analyzed using R (Version 4.1.2) statistical software to evaluate the reduction of microbial loads at each harvest, trim, and further process area. Indicator counts were converted to LogCFU/mL (g) and *Salmonella* counts were reported as LogCFU/sample. For AC, a one-way ANOVA was performed, comparing counts at each of the sampling locations, followed by pairwise multiple comparison *T*-tests, adjusted by the Benjamini–Hochberg method. If parametric assumptions were not met, the Kruskal–Wallis Test was used as a non-parametric alternative for the ANOVA, in the combination of a pairwise multiple comparison Wilcoxon’s Test adjusted by the Benjamini–Hochberg method [23]. This case was applied for EB and EC counts and *Salmonella* counts due to their low levels in several of the sampling locations. A *p*-value of 0.05 or less was selected prior to the analysis to determine significant differences.

For each of the sampling locations and for all indicators, the methodology of Shewhart’s control charts for variable data was applied to estimate statistical process control (SPC) parameters [24,25,26]. An $\bar{X}$ chart uses the average of values from a sample set to monitor process variation [26]. The $\bar{X}$ chart was developed using the central line, grand average, or overall mean count ($\overset{=}{X}$), Lower Control Limit (LCL), and Upper Control Limit (UCL). The $\overset{=}{X}$ parameter was calculated using an average per week (*n* = 5 samples) followed by the average of the 10 weeks’ means for each of the sampling locations in the pork processing line. The LCL and UCL parameters were estimated using equations 1 and 2, where $\bar{s}$ is the average standard deviation and *A*_3_ (1.427) is a standard factor value based on 5 samples collected each week during a 10-week period [25,26]. A similar SPC process was implemented in a beef bio-mapping study by Vargas et al. in 2022 [27].
(1)$$Lower\;Control\;Limit\;\left(LCL\right)=\overset{=}{X}-A_{3}\bar{s}$$
(2)$$Upper\;Control\;Limit\;\left(UCL\right)=\overset{=}{X}+A_{3}\bar{s}$$

## 3. Results

The LogCFU/mL or LogCFU/g counts for all indicators showed significant differences between sampling locations specifically at the harvest and further process stages. For *Salmonella* counts, results are presented in LogCFU/sample using a 50 mL or 50-g sample basis. The reduction at harvest (from gambrel table to after PAA cabinet location) was on average 2.27 Log_10_CFU/mL for AC (*p*-value < 0.001), 2.46 Log_10_CFU/mL for EB (*p*-value < 0.001), and 2.24 Log_10_CFU/mL for EC (*p*-value < 0.001) counts. Trim sample indicator counts fluctuated based on cut. The highest AC for trim was found at neck trim with an average of 2.83 LogCFU/g. Trim results for EB counts revealed that loin trim (1.08 LogCFU/g) and boneless picnic (0.97 LogCFU/g) showed the highest values. The highest EC count for trim was found at loin trim (0.80 LogCFU/g). 

### 3.1. Aerobic Counts (AC)

The grand average AC levels observed in the bio-mapping, standard deviation, and upper and lower control limits for statistical process control are presented in Table 1. The average AC at the gambrel table was 5.69 LogCFU/mL, and AC reduction at harvest was continuous from one sampling location to the next. ANOVA results indicate significant differences between gambrel table, the subsequent harvest locations (after polisher, before final rinse, after final rinse) and the last two locations (post snap chill and after PAA cabinet) (*p*-value < 0.001). There were no significant differences between after polisher, before final rinse and after final rinse locations (*p*-value = 0.23). Similarly, there were no significant differences between the post snap chill and after PAA cabinet locations (*p*-value = 0.19). For trim samples, no significant differences were observed between the four types of trim analyzed (*p*-value = 0.052). Further process samples showed higher AC for sausage samples (5.28 LogCFU/g). AC levels for AMR, brick, and sausage had significant differences with each other (*p*-value < 0.001). The average AC for AMR and brick locations were 3.86 and 3.11 LogCFU/g, respectively, as displayed in Figure 2.

### 3.2. Enterobacteriaceae (EB) Counts

For EB counts, detailed in Table 2, a non-parametric approach test was used for analysis due to low counts in late harvest steps and trim samples. These tests do not follow any specific distribution as in the case of parametric tests. When assumptions were not met for performing ANOVA, a Kruskal–Wallis test was applied to find differences between sampling locations at each of the processing stages. At the harvest floor, EB counts were significantly different between the gambrel table and before and after final rinse locations (*p*-value < 0.001). Furthermore, statistically significant differences were found between gambrel table, after polisher and the last two sampling locations at harvest (post snap chill and after PAA cabinet) (*p*-value < 0.001). For trim, the highest average EB count was at loin trim (0.80 LogCFU/g). Kruskal–Wallis test analysis shows that there are no significant differences between trim sampling locations (*p*-value = 0.269). The average EB counts for further process were 3.19, 2.60, and 1.27 LogCFU/g for sausage, AMR, and brick, respectively. No significant differences were found between sausage and AMR EB counts (*p*-value = 0.13), as shown on the bio-map in Figure 3.

### 3.3. Generic E. coli (EC) Counts

Similar to the case for EB counts, a non-parametric approach with Kruskal–Wallis test was used for analysis of EC counts, summarized in Table 3. The prevalence of EC in harvest stage samples analyzed was 100% in gambrel table, 50% in after polisher, 88% in before final rinse, 92% in after final rinse, 32% in post snap chill, and 22% in after PAA cabinet. For trim samples, the prevalence of EC was 16% for boneless picnic, 28% for belly trim, 18% for neck trim, and 24% for loin trim. At the further process stage, the prevalence of EC was 80% for AMR, 14% for brick, and 74% for sausage. The average EC count at the gambrel table location was 3.11 LogCFU/mL. The overall reduction in EC counts at harvest was 2.24 LogCFU/mL. There were no statistically significant differences between before final rinse and after final rinse locations (*p*-value = 0.32). This also applied for the post snap chill and after PAA cabinet locations (*p*-value = 0.24), with average values of 0.97 and 0.87 LogCFU/mL, respectively. Trim EC counts followed the trend observed at the last location analyzed at the harvest stage with the highest count found at loin trim (0.80 LogCFU/g). There were no statistically significant differences in EC counts between the four types of trim evaluated (*p*-value = 0.363). The highest EC count for further process samples was sausage links (1.60 LogCFU/g). The average EC counts showed statistically significant differences (*p*-value < 0.001) between the three sampling locations in this stage as displayed on the bio-map in Figure 4.

### 3.4. Salmonella Detection and Enumeration

*Salmonella* counts were very low when analyzed on a per-mL or per-g basis in most samples, thus when transformed to LogCFU/mL and LogCFU/g, counts resulted in negative values, making analysis and visualization more difficult for interpretation. Thereby, all data were transformed to LogCFU/sample which is equivalent to LogCFU/50 mL (harvest swabs) and LogCFU/50 g (trim and ground samples) to facilitate data interpretation. This transformation to LogCFU/sample has also been applied in other studies [28,29]. The Limit of Quantification (LOQ) for SalQuant™ on pork carcass swabs is 1 CFU/mL. The LOQ for SalQuant™ on pork trim and ground pork is 1 CFU/g. When using SalQuant™, counts can be extrapolated below the LOQ since counts are obtained from a regression equation specific to each matrix, provided by the methodology. To better accommodate for this extrapolation process, a new LOQ was established as 10% of the real LOQ, 0.1 CFU/mL and 0.1 CFU/g or 0.70 LogCFU/sample. Samples with a value of <0.70 LogCFU/sample were reported as 50% of the new LOQ (0.35 LogCFU/sample), which will be referred to as the new Limit of Detection (LOD). This value was also used for samples that were not quantifiable during the 6 and 7-h timepoints but found positive for *Salmonella* prevalence. Samples that were not quantifiable nor detected were reported as 0 logCFU/sample. A summary of the conditions and parameters used can be found in Table 4 for carcass swabs, and Table 5 for trim and ground pork. A similar approach was used by De Villena et al. to present poultry rinsates *Salmonella* quantification and prevalence data together [30].

For *Salmonella* prevalence results, a total of 125/650 samples were found positive (19%). Prevalence was evaluated using BAX^®^ System Real-Time Salmonella assay. Table 6 shows *Salmonella* prevalence at each sampling location throughout the pork processing line. From those *Salmonella*-positive samples, 47 samples (37%) were suitable for enumeration using the BAX^®^ System SalQuant™, the majority detected at the gambrel table location, as shown on the bio-map in Figure 5. From these enumerable samples, 60% were within 0.97–1.97 LogCFU/sample and the remaining portion (40%) were in the 2.00–4.02 LogCFU/sample range. The average *Salmonella* load at the gambrel table location was 1.87 LogCFU/sample. Other harvest locations with SalQuant™ results include after polisher (1.77 LogCFU/sample), before final rinse (2.07 LogCFU/sample), and after final rinse (2.13 LogCFU/sample). A non-parametric approach with Kruskal–Wallis test was used for analysis of *Salmonella* results. A significant difference in counts was found between the gambrel table and after final rinse (*p*-value < 0.001). No significant differences were found between before and after final rinse locations (*p*-value = 0.07). Furthermore, no significant differences were found between the after polisher and before final rinse locations (*p*-value = 0.15). The post snap chill and after PAA cabinet locations showed no significant differences (*p*-value = 0.80). These two latter locations were significantly different when compared to the gambrel table (*p*-value < 0.001). For trim results, neck trim and boneless picnic yield the highest mean *Salmonella* counts with 2.53 and 2.26 LogCFU/sample, respectively. Neck trim and boneless picnic *Salmonella* levels showed no significant differences (*p*-value = 0.94). Nevertheless, these two locations were significantly different when compared to belly trim and loin trim cuts (*p*-value = 0.001). It is important to note that all loin trim samples were estimated at 0.0 LogCFU/sample, which means no *Salmonella*-positive samples were found at this location accounting both SalQuant™ and prevalence results. *Salmonella* enumeration results at the further process stage indicate that brick samples yield the highest mean with 2.27 LogCFU/sample. *Salmonella* prevalence from highest to lowest is the following: brick, sausage, and AMR. There were no significant differences between sampling locations after Kruskal–Wallis test analysis (*p*-value = 0.137).

## 4. Discussion

Microbial indicator results observed in the present study suggest significant reductions in microbial contamination at most of the sampling locations as the process moves forward at the harvest stage. Despite these reductions, an increase in counts in indicator organisms occurs at the further process stage, more substantial for AC and EB counts. Based on the design of the study, the gambrel table location can be used as a measure of the incoming load as it constitutes the first sampling point in the process. This is not a true incoming load data point, since the sampling takes place after stunning, bleeding, scalding, and dehairing processes, all with potential bacterial reduction effects. The gambrel table represents a post-scalding point which means carcasses have been exposed to scalding water at 145 °F (62 °C) for 5 min. During the scalding process, a counter current type is recommended to increase heating efficiency and water cleanliness as a result of fresh or recirculated scald water that flows into the scalder in opposite direction from the carcasses [31]. The reduction of indicator organisms at harvest is apparent with 2.27 LogCFU/mL for AC, 2.46 LogCFU/mL for EB, and 2.24 LogCFU/mL for EC counts. These reductions can be attributed to the in-plant processing controls in place. A comprehensive visual analysis of indicator counts in Figure 2, Figure 3 and Figure 4 seems to show that counts follow a U-shape pattern from the first sampling location at the harvest stage to the last product type in further process stages.

For *Salmonella* enumeration, the BAX System SalQuant™ detected the majority of *Salmonella*-positive samples at gambrel table (20/34 samples), after final rinse (9/16 samples), before final rinse (6/9 samples), and boneless picnic (5/12 samples) locations. These values are the result of *Salmonella* being detected at the 6-h recovery timepoint for both harvest swabs and trim cuts. Furthermore, these results serve as a way to highlight the value of pathogen enumeration over just prevalence data. *Salmonella* quantification provides a variety of advantages to assess process performance and for final product assessment. With *Salmonella* quantification, processors obtain precise information in regard to *Salmonella* location and level of contamination, which supports tracking from incoming lots to the final product. In the case of the final product, the SalQuant™ methodology, as implemented in this study, can facilitate hold and release decisions in ground products on the basis of lower contamination levels and provide faster results to make diversion decisions to lower risk product pipelines. Moreover, results from *Salmonella* quantification offer a better assessment of the true meaning of *Salmonella* prevalence. Prevalence results provide limited information as they are based on presence or absence, whereas quantification yields an estimate of *Salmonella* level on the positive samples. The presence of *Salmonella* as detected by full enrichment methodologies does not necessarily translate into the possibility to cause illness as the minimum infectious dose of *Salmonella* is estimated at 10^4^ CFU [32,33]. *Salmonella* performance standards in poultry products and the proposed for pork products are currently based on *Salmonella* prevalence. The prevalence-based approach unfortunately misses the full risk management aspect for foodborne illnesses caused by *Salmonella* For instance, the risk resulting from 5 LogCFU/sample of *Salmonella* within a pork product to that in a product with 0.5 LogCFU/g is significantly different. In these cases, *Salmonella* enumeration provides more relevant and critical information that cannot be obtained with prevalence data alone. 

The methods herewith implemented with this 10-week baseline and bio-mapping study show that microbial contamination varies progressively during the pork processing stages. As emphasized by De Villena et al., bio-mapping baselines with pathogen quantification can facilitate the development of statistical process control parameters to support food safety management decision-making, in this case in pork operations. The value of bio-mapping studies lies in identifying where food safety risk is the greatest on the processing line and taking actionable data-driven and science-based decisions for continuous improvement. Moreover, the use of nonparametric statistical process control can help data management in terms of using *Salmonella* prevalence and quantification data together, which may further improve the process of decision-making than the case when using only prevalence data [30]. As part of in-plant process verification testing, with the implementation of bio-mapping, each plant can monitor their systems closely for decreases in microbial levels and ensure the highest attainable levels of microbial reductions in raw products [34]. Another advantage of plant bio-mapping is to assess the effectiveness of antimicrobial intervention schemes. Bio-mapping studies serve as a measure of the microbial recovery at pre- and post-intervention stages for the whole process [35]. Bio-mapping not only represents a systematic analysis of individual hurdles within the whole system, but profiling results indicate where intervention strategies are working effectively or failing. Moreover, there are situations in which incidents of contamination are not the result of the process itself and may be tied to process management or a lack of understanding of how each part of the process affects the entire system [36]. In these cases, bio-mapping or process mapping is very useful in providing an establishment with ongoing information about process performance. 

Bio-mapping results can be used to generate SPC on microbial counts data and serve as scientific support for food safety management decisions. As explained in a USDA-FSIS guide on sampling requirements to demonstrate process control in slaughter operations, SPC involves the use of statistical methods, such as control charts, to evaluate the variability of a process [37]. The foundation for conducting SPC is to maintain control despite the intrinsic variability of the process and improve the performance at strategic locations to improve microbial levels in the final product. One application at the harvest stage is to measure and manage the microbiological contamination of carcasses. In addition, SPC is a powerful tool for establishments to monitor and interpret data collected for ongoing HACCP verification. It can also provide an early warning if their process is not functioning as designed or if it is trending towards failure. Consequently, this warning serves as an opportunity for establishments to make process modifications to control deviations without failing the desired performance objectives. The Upper Control Limit (UCL) and Lower Control Limit (LCL) values have specific interpretations and must be used accordingly on a plant-to-plant basis. Vargas et al. explains that the LCL is not really observations out of control since lower counts than the lower limit translate into better microbial performance [27]. Conversely, the UCL can serve as an alert to explore occasions when the process is out of control and guide the need for root cause analysis and corrective action implementation in a timely manner to avoid more deviations within the process.

### 4.1. Aerobic Counts (AC)

AC levels observed at the harvest stage show a statistically significant 1.64 LogCFU/mL reduction until before the final rinse location (*p*-value < 0.05). The subsequent location, after the final rinse, increases in AC but a 1.60 LogCFU/mL reduction is observed at the post snap chill location. Hong et al. 2008 implemented the use of aerobic plate counts as a measure of HACCP’s effectiveness in a pork processing facility with a focus on chilling, cutting, and packing steps [38]. The mean aerobic plate counts from swabs at chilling and cutting after HACCP implementation throughout the course of the four-year duration of the study were 1.80 and 2.36 LogCFU/cm^2^, respectively. The mean AC in the current study at the post snap chill is 2.92 LogCFU/mL, which is higher than Hong et al.’s 2008 findings, but they also pointed out that is not easy to compare meat plants as there are too many variables that need to be accounted for on a per facility basis [38]. For trim, the highest mean count was found at neck trim with 2.83 LogCFU/g, a comparative value to the cutting aerobic plate count in the study by Hong et al. in 2008. Several studies have evaluated AC in the pork processing line using the swab technique and traditional plate count methods [38,39,40,41,42] or 3M Petrifilm™ [43] for microbial analysis. Ba et al. in 2019 included not only sampling of carcasses at different slaughter stages but also on retail cuts (24 h post-mortem), comprising neck trim, loin trim, and butt trim. The aerobic plate counts in this study on the carcass surface at different slaughter stages (after bleeding and evisceration) show reductions throughout the process. At the time of slaughter, AC was 7.03 LogCFU/100 cm^2^ and these levels were reduced to 6.32 LogCFU/100 cm^2^. This reduction may be associated with the synergistic effects of scalding with hot water and singeing during the slaughter process [43]. Furthermore, these results are in agreement with the findings by Pearce et al. in 2004 and Spescha et al. in 2006 in terms of the role of the scalding process in aerobic bacteria reduction [40,42]. Despite the fact that the mean AC for trim cuts in this study range from 0.93 to 0.97 LogCFU/g and Ba et al., trim counts were estimated at 1.57 LogCFU/100 cm^2^, an overall AC reduction is observed in both studies from harvest to primal cuts [43]. The variability in AC seen at the further process stage is somewhat concerning, since this involves the final product. A series of studies as part of the National Pork Retail Microbiological Baseline, published by the National Pork Board and American Meat Science Association were conducted to further elucidate such findings [44]. Studies to determine the presence of indicator microorganisms on pork carcasses provide insightful information regarding the population of microorganisms that may be found on pork resulting from cross-contamination or poor handling/processing practices. There is value in the assessment of the final product as pork can be re-contaminated with bacteria during fabrication, packaging, and in later stages of the retail chain. The mean AC (*n* = 40) of three types of pork plants processing ground pork and/or sausage were the following: 3.3 LogCFU/g (slaughtering and fabricating plants), 3.0 LogCFU/g (further-processing plants) and 2.9 LogCFU/g (hot-boning sausage plants) [44]. Statistical significant differences were found when comparing the results from slaughtering and fabricating plants to the other plant types (*p*-value < 0.05) [44]. These findings support the results for brick average AC of 3.11 LogCFU/g observed in this study. The spike in mean AC for sausage (5.28 LogCFU/g) may be related to the process of creating ground products and other potential sources of cross-contamination, such as glands [44]. Interestingly, the AC levels for sausage are aggregated in three separate regions of the box plot (top, center, and bottom) which can be correlated to samples from weeks with lower and higher AC. For instance, AC results for weeks 1, 5 and 8 are on the lower 2–4 LogCFU/g range, whereas weeks 7, 9 and 10 values fall on the 7, 8 logCFU/g range. This observation affirms the value of conducting in-plant baseline studies over extended periods of time which can better asssess a plant’s food safety system and the need for interventions at specific points throughout the pork processing line. 

### 4.2. Enterobacteriaceae (EB) Counts

EB counts at the harvest floor show similar trends to those observed for AC. A significant 1.71 LogCFU/mL reduction is obtained from the gambrel table to the after polisher location (*p*-value < 0.05). The before and after final rinse locations result in no statistically significant differences between each other, but further EB count reductions are obtained at the post snap chill and after PAA cabinet locations. This reduction of EB in chilling is in accordance with the findings of four prior studies [42,45,46,47]. The study by Lenahan et al. in 2009 reported significant differences (*p*-value < 0.05) before and after chilling in carcass counts after using the nonparametric Kruskal–Wallis test for analysis [47]. Barco et al. in 2014 noted in a review paper comprising work supported by European Food Safety Authority (EFSA) that chilling efficacy depends on the plant’s specific process, study design, and sampling locations [48]. Furthermore, the potential of reducing microbial contamination is based on the counts of locations prior to the chilling stage. Subsequent steps from the polishing step through the post-chill step resulted in a continous reduction of EB counts in a study by Corbellini et al. in 2016, which involved an assessment of the effect of slaughterhouse and sample day to establish a relationship between EB counts and the probability of a carcass being *Salmonella*-positive [49]. Their findings showed a 3.6 log reduction in EB counts post-scalding and an estimated 1.7 LogCFU/cm^2^ after polishing. In the present study, EB average counts at the after polisher step was 1.84 LogCFU/mL, which correlates to Corbellini et al.’s findings [49]. EB counts remain constant within the trim stage for all four cuts, with the highest average value seen at loin trim (1.08 LogCFU/g). Specha et al. in 2006 reported an increase in EB counts after trimming in neck, belly, and back pork cuts in which neck cuts showed higher counts [42]. Similar to the trend observed with AC, EB counts for the further process stage showed an increase in counts, with the highest value found in sausage links (3.19 LogCFU/g).

### 4.3. Generic E. coli (EC) Counts

There is a significant 2.13 LogCFU/mL EC reduction from the gambrel table to the after polisher location (*p*-value < 0.05). This is in accordance with prior reported studies [39,50,51]. Namwar & Warriner found that EC levels were significantly reduced after polishing and triple singeing. EC was recovered sporadically on eviscerated carcasses at later stages with several samples negative for this indicator organism. They also recovered EC at low levels from the rinse water, which is also observed in this study with an increase in EC counts at the rinse locations. One of the highlights from the review by Belluco et al. in 2015 was that scalding and chilling processes are effective in reducing EC on pig carcasses [51]. Furthermore, other studies have reported EC count reductions after the chilling process [52,53]. Ba et al. in 2019 estimated EC counts at bleeding (3.63 LogCFU/100 cm^2^), evisceration (3.43 LogCFU/100 cm^2^), and on carcass surfaces with 4% lactic acid spray (1.32 LogCFU/100 cm^2^), showing a statistically significant reduction throughout the pork processing line [43]. The samples post-spray treatment encompasssed sampling of neck trim, loin trim, and butt trim. These findings correlate to what was observed in this study with the EC counts from slaughter to trim stages. EC counts for further process samples showed a similar trend to the results from aerobic and EB counts, in which sausage presented the highest count with an average value of 1.60 LogCFU/g. Nevertheless, brick values were significantly lower with a mean count of 0.71 LogCFU/g. These results are similar to the EC counts in ground pork and sausage reported in the National Pork Retail Microbiological Baseline with estimates for hot-boning sausage plants, slaughtering and fabricating plants, and further processing plants at 1.1, 1.0, and 0.9 LogCFU/g, respectively [44]. 

### 4.4. Salmonella

Although a series of studies have been conducted to estimate *Salmonella* prevalence in pork [40,41,49,54,55,56,57,58], only a few have evaluated *Salmonella* enumeration [43,45]. *Salmonella* prevalence results in the present study show 68% at the gambrel table location, which constitutes the first sampling location at the harvest stage, and 2% at the end of this stage (after PAA cabinet). There is variability in *Salmonella* prevalence for the trim cuts analyzed with the highest values observed in neck trim (26%) and boneless picnic (24%). Ground product *Salmonella* prevalence was estimated at 28% and 24% for Brick and Sausage, respectively. Algino et al.’s findings estimated *Salmonella* prevalence in unskinned carcasses (*n* = 121) during the prewash stage at 8.26% whereas post-chill yield 18.18% [55]. Weissman and Carpenter suggested that reasons for variance in *Salmonella* prevalence on hog carcasses may be due to differences in herd incidence, transportation and handling methods, plant processing schemes, and bacteriological methodologies [59]. Botteldoorn et al. estimated *Salmonella* prevalence before chilling (*n* = 370) at 37%, which is similar to the 32% *Salmonella* prevalence in the after final rinse location evaluated in the present study, which is the location assessed prior to the chilling step [57]. In a study by Pearce et al., *Salmonella* was detected on 31% of carcasses after bleeding, 1% after scalding, 7% after dehairing, 0% after polishing, and 7% after evisceration [40]. This reduction in *Salmonella* may be linked to the effect of scalding to reduce indicator organisms and the incidence of pathogens. Corbellini et al. found that *Salmonella* prevalence followed the trend of EB concentration during slaughter steps [49]. This also can be seen in the present study when comparing Figure 3 which displays EB counts and Figure 5 with *Salmonella* counts and prevalence. The similarities between EB counts and *Salmonella* prevalence are notorious in the harvest stages. The sampling locations with the highest mean EB counts in descending order were gambrel table, after final rinse, and before final rinse. Similarly, *Salmonella* incidence is highest at the gambrel table (68%), followed by after final rinse (32%) and before final rinse (18%). Based on these results, EB may represent a good hygienic indicator for harvesting stages at this processing facility. *Salmonella* prevalence was estimated at 38% for sausage samples in a study by Weissman and Carpenter [59]. In the current study, the incidence of *Salmonella* in sausages was 24%. The National Pork Retail Microbiological Baseline estimated *Salmonella* incidence in ground pork and/or sausage in hot boning plants (10%), slaughtering/fabricating plants (7.5%), and further processing plants (0%) [44]. In addition, pre-packaged ground pork (*n* = 96) was linked to 12.5% *Salmonella* prevalence when including retail sample results [44].

Despite the reduction seen in *Salmonella* enumeration results between the gambrel table and after polisher locations, an increase is subsequently seen in the before and after final rinse locations with average counts of 2.07 and 2.13 LogCFU/sample, respectively. *Salmonella* was undetected at the post snap chill location. After the PAA cabinet treatment, 1 sample was found positive at the LOD. Duggan et al. in 2010 conducted *Salmonella* enumeration by MPN technique in order to identify points of *Salmonella* contamination from lairage to the time the carcass was processed in the boning hall [45]. Their results showed a correlation between the presence of *Salmonella* and higher EB levels for all sample types (pre-chill and post-chill carcasses, pork cuts). Furthermore, *Salmonella* enumeration results were relatively low with median values of 0.009 MPN/cm^2^, 0.0075 MPN/cm^2^, and 0.3 MPN/g for pre-chill carcasses, post-chill carcasses and pork cuts, respectively. Ba et al. in 2019 estimated *Salmonella enterica* at 4.50, 3.59, and 0.69 LogCFU/100 cm^2^; after bleeding, evisceration, and on carcass surfaces with 4% lactic acid spray [43]. This study also assessed *Salmonella* on retail’s cut surfaces but results are presented as not detected or insufficient data for calculation. Corbellini et al. noted that when comparing earlier stages during slaughter up until the post-chill stage, a decrease in the number of *Salmonella*-contaminated carcasses may indicate that process controls were effective in reducing the level of contamination along with the implementation of good hygiene practices [49]. The increase in *Salmonella* levels observed in ground products can potentially be the result of more extensive handling and processing. Other potential reasons why there is an increase in *Salmonella* levels in the ground product when low levels and absence has been observed at the end of harvest are related to the nature and type of ground product, the meat source(s) used during the grinding process, grinding of pork trim to expose more surface area, and extraintestinal tissues sources as potential sources of *Salmonella*. Extraintestinal tissues include liver, spleen, tonsils, cervical, inguinal, and mesenteric lymph nodes, among others. *Salmonella* was recovered and isolated in at least 1 tissue of 6 selected tissues (serum samples, cecal contents, mesenteric lymph nodes, carcass swabs, liver, and spleen) evaluated in 226 out of the 442 total sampled pigs [60]. More specifically, the results showed that *Salmonella* Typhimurium was present in a large amount of cecal contents and mesenteric lymph nodes from both groups of carcasses assessed (with or without a history of clinical salmonellosis). In the current plant, boneless picnic is ground as one of the components to form ground brick and sausage links. One study on the simulation of *Salmonella* Typhimurium DT104 transfer during pork grinding highlighted how *Salmonella* present on a single piece of meat may be transferred to several portions of minced meat due to cross-contamination in the grinder [61].

## 5. Conclusions

Pathogen quantification in pork processing settings may be limited due to difficulties with pathogen recovery, stressed conditions for pathogens exposed to different processing operations and antimicrobial interventions. The quantification techniques implemented in this study have shown to recover pathogens from positive samples as a result of a recovery stage with short enrichment steps that increase the likelihood of collecting quantification estimation data. In addition, *Salmonella* quantification may constitute an advantage for risk management as results can guide decisions on the basis of load and level in specific stages in the process, rather than just presence or absence. Quantification can benefit the pork industry in several ways from live production (predict pathogen load, influence slaughter order, adjust interventions), to processing (assess intervention efficacy, corrective action responses) to the final product (product release decisions and consumer health risk).

In this study, a quantitative microbial bio-map was developed on the harvest floor and final products in a large USDA-inspected and HIMP pork processing plant after NSIS implementation. This study provides evidence for the application of emerging technologies for pathogen quantification and indicator levels in pork samples. It also serves as the basis for developing statistical process control variables based on process bio-mapping baselines at different stages during processing to support food safety management decision-making for controlling pathogens in pork products and guide process changes and speed line modifications.

This study shows that microbial indicators and *Salmonella* levels were reduced throughout the pork harvest floor, demonstrating significant process control with reductions occurring at the after PAA cabinet location in whole carcass samples following the specified sampling protocols. Nevertheless, for trim and ground sample processing, *Salmonella* and indicator microorganisms were not only detected, but also quantifiable albeit at low numbers at certain points indicating the need for additional interventions after carcass processing and further evaluation of potential sources of contamination during fabrication and further processing. The utilization of a rapid PCR-based enumerative method for pathogens, in conjunction with indicator levels, provides the pork industry with a tool for data-driven decisions throughout the pork supply chain to target points in the process of greatest concern, establish statistical process control thresholds, comply with incoming performance standards, and to mitigate the risk to public health of foodborne illness.

## Figures and Tables

**Figure 1 foods-11-02580-f001:**
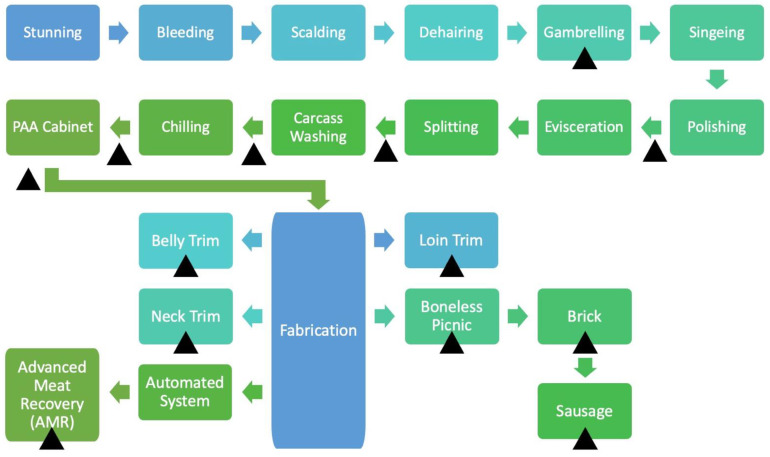
Schematic representation of the pork processing line. Sampling locations are indicated by solid triangles.

**Figure 2 foods-11-02580-f002:**
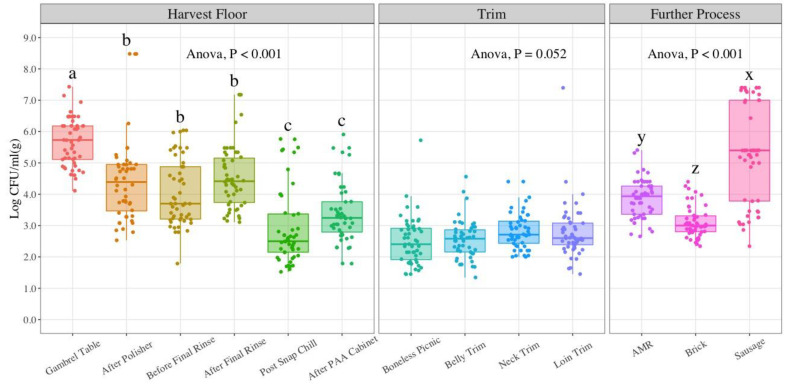
Aerobic counts in LogCFU/mL (g) at each processing stage throughout the pork processing line (*n* = 50; N = 650). The limit of detection (LOD) at harvest is 6 CFU/mL (0.78 LogCFU/mL); for trim and further process samples 5 CFU/g (0.70 LogCFU/g). In the boxplot, the horizontal line crossing the box represents the median, the top and bottom lines represent the lower (0.25) and upper (0.75) quartiles, the vertical top lines represent 1.5 times the interquartile range and the vertical bottom line represents 1.5 times the lower interquartile range. The dots represent actual data points. Box plots with different letters (a–c); (x–z) represent statistical significant differences acccording to ANOVA analysis followed by a pairwise T-test comparison using a *p*-value < 0.05. PAA (Peroxyacetic acid); AMR (Advanced Meat Recovery).

**Figure 3 foods-11-02580-f003:**
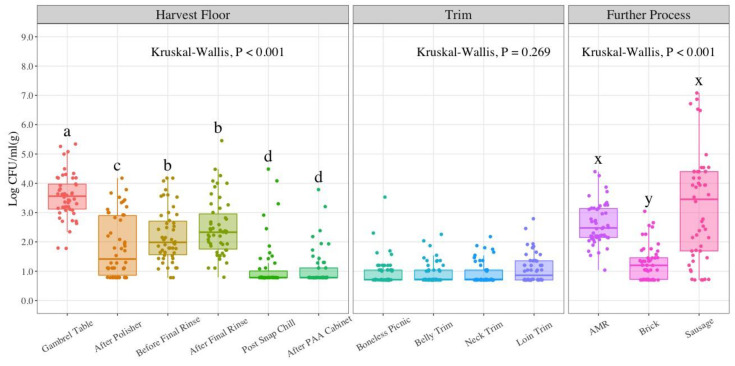
Enterobacteriaceae (EB) counts in LogCFU/mL (g) at each processing stage throughout the pork processing line (*n* = 50; N = 650). The LOD at harvest is 6 CFU/mL (0.78 LogCFU/mL); for trim and further process samples 5 CFU/g (0.70 LogCFU/g). In the boxplot, the horizontal line crossing the box represents the median, the top and bottom lines represent the lower (0.25) and upper (0.75) quartiles, the vertical top lines represent 1.5 times the interquartile range and the vertical bottom line represents 1.5 times the lower interquartile range. The dots represent actual data points. Box plots with different letters (a–d); (x, y) represent statistical significant differences according to Kruskal–Wallis analysis followed by pairwise comparison with Wilcoxon’s test at *p*-value < 0.05. PAA (Peroxyacetic acid); AMR (Advanced Meat Recovery).

**Figure 4 foods-11-02580-f004:**
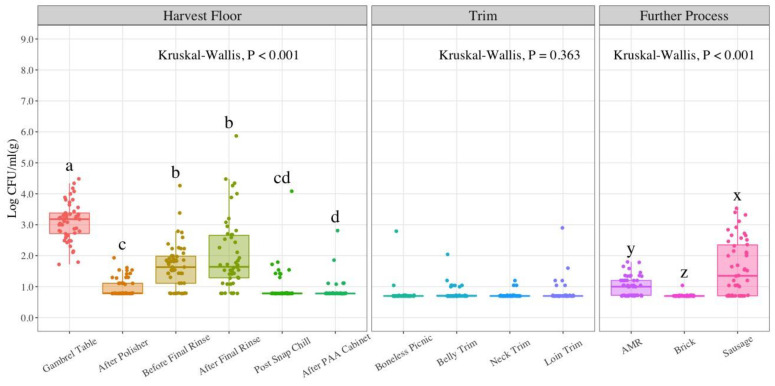
Generic *E. coli* (EC) counts in LogCFU/mL (g) at each processing stage throughout the pork processing line (*n* = 50; N = 650). The LOD at harvest is 6 CFU/mL (0.78 LogCFU/mL); for trim and further process samples 5 CFU/g (0.70 LogCFU/g). In the boxplot, the horizontal line crossing the box represents the median, the top and bottom lines represent the lower (0.25) and upper (0.75) quartiles, the vertical top lines represent 1.5 times the interquartile range and the vertical bottom line represents 1.5 times the lower interquartile range. The dots represent actual data points. Box plots with different letters (a–d); (x–z) represent statistical significant differences according to Kruskal–Wallis analysis followed by pairwise comparison with Wilcoxon’s test at *p*-value < 0.05. PAA (Peroxyacetic acid); AMR (Advanced Meat Recovery).

**Figure 5 foods-11-02580-f005:**
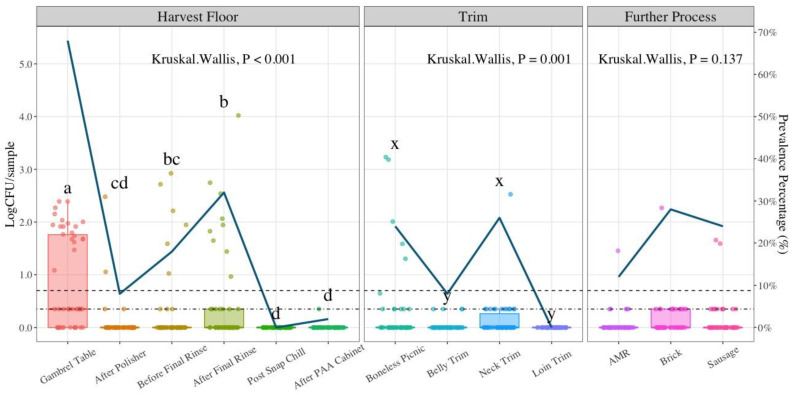
*Salmonella* counts (LogCFU/sample), left Y axis, and prevalence (solid blue line), right Y axis, at each processing stage throughout the pork processing line. In the boxplot, the horizontal line crossing the box represents the median, the top and bottom lines represent the lower (0.25) and upper (0.75) quartiles, the vertical top lines represent 1.5 times the interquartile range and the vertical bottom line represents 1.5 times the lower interquartile range. The dots represent actual data points. The LOQ for SalQuant™ on pork carcass swabs is 1 CFU/mL. The LOQ for SalQuant™ on pork trim and ground pork is 1 CFU/g. The new LOQ is represented by the dashed line. The new LOD is represented by the dot-dash line. This LOD applies for samples that were Quant Negative but positive after prevalence testing. Box plots with different letters (a–d); (x, y) represent statistical significant differences according to Kruskal–Wallis analysis followed by pairwise comparison with Wilcoxon’s test at *p*-value < 0.05. PAA (Peroxyacetic acid); AMR (Advanced Meat Recovery).

**Table 1 foods-11-02580-t001:** Statistical process control parameters for aerobic counts calculated at each sampling location (*n* = 5 samples/location) collected during 10 weeks throughout the pork processing line.

Sampling Location	Processing Stage	$\bar{X}\;\pm \;\sigma$	Log CFU/mL(g) LCL ^1^	Log CFU/mL(g) UCL ^2^
Gambrel Table	Harvest	5.69 ± 0.77 ^a^	4.60	6.78
After Polisher	Harvest	4.45 ± 1.35 ^b^	2.52	6.38
Before Final Rinse	Harvest	4.05 ± 1.07 ^b^	2.53	5.57
After Final Rinse	Harvest	4.52 ± 0.99 ^b^	3.11	5.93
Post Snap Chill	Harvest	2.92 ± 1.20 ^c^	1.20	4.64
After PAA ^3^ Cabinet	Harvest	3.42 ± 0.96 ^c^	2.05	4.79
Boneless Picnic	Trim	2.51 ± 0.77	1.41	3.61
Belly Trim	Trim	2.55 ± 0.62	1.67	3.43
Neck Trim	Trim	2.83 ± 0.58	2.01	3.65
Loin Trim	Trim	2.80 ± 0.89	1.53	4.07
AMR ^4^	Further Process	3.86 ± 0.63 ^y^	2.97	4.75
Brick	Further Process	3.11 ± 0.50 ^z^	2.39	3.83
Sausage	Further Process	5.28 ± 1.58 ^x^	3.02	7.54

$\bar{X}$ = expected average value, σ = average standard deviation of the mean ^1^ LCL = Lower control limit; ^2^ UCL = Upper control limit; ^3^ PAA = Peroxyacetic acid; ^4^ AMR = Advanced Meat Recovery. Values with different letters (a–c); (x–z) represent significant statistical differences (*p*-value < 0.05).

**Table 2 foods-11-02580-t002:** Statistical process control parameters for Enterobacteriaceae (EB) counts calculated at each sampling location (*n* = 5 samples/location) collected during 10 weeks throughout the pork processing line.

Sampling Location	Processing Stage	$\bar{X}\pm \sigma$	Log CFU/mL(g) LCL ^1^	Log CFU/mL(g) UCL ^2^
Gambrel Table	Harvest	3.55 ± 0.77 ^a^	2.46	4.64
After Polisher	Harvest	1.84 ± 1.05 ^c^	0.35	3.33
Before Final Rinse	Harvest	2.21 ± 0.91 ^b^	0.91	3.51
After Final Rinse	Harvest	2.47 ± 0.99 ^b^	1.06	3.88
Post Snap Chill	Harvest	1.15 ± 0.85 ^d^	0.00	2.36
After PAA ^3^ Cabinet	Harvest	1.09 ± 0.64 ^d^	0.17	2.01
Boneless Picnic	Trim	0.97 ± 0.50	0.26	1.68
Belly Trim	Trim	0.93 ± 0.38	0.39	1.47
Neck Trim	Trim	0.93 ± 0.39	0.37	1.49
Loin Trim	Trim	1.08 ± 0.50	0.37	1.79
AMR ^4^	Further Process	2.60 ± 0.70 ^x^	1.60	3.60
Brick	Further Process	1.27 ± 0.60 ^y^	0.41	2.13
Sausage	Further Process	3.19 ± 1.80 ^x^	0.62	5.76

$\bar{X}$ = expected average value, σ = average standard deviation of the mean ^1^ LCL = Lower control limit; ^2^ UCL = Upper control limit; ^3^ PAA = Peroxyacetic acid; ^4^ AMR = Advanced Meat Recovery. Values with different letters (a–d); (x, y) represent significant statistical differences (*p*-value < 0.05).

**Table 3 foods-11-02580-t003:** Statistical process control parameters for generic *E. coli* (EC) counts calculated at each sampling location (*n* = 5 samples/location) collected during 10 weeks throughout the pork processing line.

Sampling Location	Processing Stage	$\bar{X}\pm \sigma$	Log CFU/mL(g) LCL ^1^	Log CFU/mL(g) UCL ^2^
Gambrel Table	Harvest	3.11 ± 0.62 ^a^	2.23	3.99
After Polisher	Harvest	0.98 ± 0.30 ^c^	0.55	1.41
Before Final Rinse	Harvest	1.67 ± 0.72 ^b^	0.64	2.70
After Final Rinse	Harvest	2.03 ± 1.14 ^b^	0.40	3.66
Post Snap Chill	Harvest	0.97 ± 0.53 ^d^	0.21	1.73
After PAA ^3^ Cabinet	Harvest	0.87 ± 0.33 ^d^	0.40	1.34
Boneless Picnic	Trim	0.75 ± 0.30	0.32	1.18
Belly Trim	Trim	0.78 ± 0.22	0.47	1.09
Neck Trim	Trim	0.73 ± 0.10	0.59	0.87
Loin Trim	Trim	0.80 ± 0.35	0.30	1.30
AMR ^4^	Further Process	1.01 ± 0.32 ^y^	0.55	1.47
Brick	Further Process	0.71 ± 0.05 ^z^	0.64	0.78
Sausage	Further Process	1.60 ± 0.90 ^x^	0.32	2.88

$\bar{X}$ = expected average value, σ = average standard deviation of the mean ^1^ LCL = Lower control limit; ^2^ UCL = Upper control limit ^3^ PAA = Peroxyacetic acid; ^4^ AMR = Advanced Meat Recovery. Values with different letters (a–d); (x–z) represent significant statistical differences (*p*-value < 0.05).

**Table 4 foods-11-02580-t004:** Observed and reported parameters established for *Salmonella* quantification and prevalence analysis for pork carcass swabs.

Observed SalQuant™ Result (LogCFU/50 mL)	Observed Prevalence Result	Reported SalQuant™ Result (LogCFU/50 mL)	Reported Prevalence Result
Quant Negative	Negative	0	Negative
Quant Negative	Positive	0.35	Positive
Less than 0.70	NA ^1^	0.35	Positive
More or equal than 0.70	NA ^1^	Observed SalQuant Result	Positive

^1^ Not applicable, as prevalence test is not necessary in samples quantified by SalQuant™.

**Table 5 foods-11-02580-t005:** Observed and reported parameters established for *Salmonella* quantification and prevalence analysis for pork trim and ground pork.

Observed SalQuant™ Result (LogCFU/50 g)	Observed Prevalence Result	Reported SalQuant™ Result (LogCFU/50 g)	Reported Prevalence Result
Quant Negative	Negative	0	Negative
Quant Negative	Positive	0.35	Positive
Less than 0.70	NA ^1^	0.35	Positive
More or equal than 0.70	NA ^1^	Observed SalQuant Result	Positive

^1^ Not applicable, as prevalence test is not necessary in samples quantified by SalQuant™.

**Table 6 foods-11-02580-t006:** *Salmonella* prevalence in % and *Salmonella*-positive sample ratio at each sampling location throughout the pork processing line.

Location	Processing Stage	Prevalence % (Positive/*n*)
Gambrel Table	Harvest	68% (34/50)
After Polisher	Harvest	8% (4/50)
Before Final Rinse	Harvest	18% (9/50)
After Final Rinse	Harvest	32% (16/50)
Post Snap Chill	Harvest	0% (0/50)
After PAA ^1^ Cabinet	Harvest	2% (1/50)
Boneless Picnic	Trim	24% (12/50)
Belly Trim	Trim	8% (4/50)
Neck Trim	Trim	26% (13/50)
Loin Trim	Trim	0% (0/50)
AMR ^2^	Further Process	12% (6/50)
Brick	Further Process	28% (14/50)
Sausage	Further Process	24% (12/50)

^1^ PAA = Peroxyacetic acid; ^2^ AMR = Advanced Meat Recovery.

## Data Availability

Data available upon request from the corresponding author. The data are not publicly available due to privacy from the pork processing partner which allowed the project to be conducted within their facility.

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
