# Peer review of "Quantitative Bio-Mapping of Salmonella and Indicator Organisms at Different Stages in a Commercial Pork Processing Facility"

_foods, 2022, doi:10.3390/foods11172580_

Round 1

Reviewer 1 Report

This is an interesting study for the microbiology control in pork facility. It may provide useful information for the industry in Salmonella control.

The followings are detailed comments:

1.      Line 34, detailed data should be provided after “table location”

2.      Introduction part, line 42 to line 126 should be shortened. The writing should focused on the theme of this manuscript. Non-related information should be simplified or deleted.

3.      Line 144, each time, five samples were collected? Please make it clear.

4.      Line 149, “PAA” should give the full name when first show up in the manuscript. The author should check the manuscript for others.

5.      Line 187-188, “TEMPO is a fully automated enumeration system based onthe principle of the 16-tube Most Probable Number (MPN) method” should be deleted.

6.      Line 195-196, incubation may change the Salmonella enumeration result? Why the author didn’t use the MPN method? Please explain?

7.      Line 208, “ul” should be “µl”.

8.      Line 238, the difference is based on statistical analysis, please show the P value?

9.      Line 255, “statistical” should be deleted, the same for line 264.

10.  In discussion, the author should discuss Salmonella, AC, EB, coli-forms, and EC data together, and their relationship, instead of separately.

Reviewer 2 Report

The objective of the manuscript is to develop a baseline of selected bacteria within a commercial pork processing facility. To my point of view, the approach is sound and all the results are statistically validated. Results obtained can be applied in quantitative microbial risk assessment studies. I suggest the authors to include a discussion regarding the probability of cross-contamination throughout the processing stages of the facility. 

Reviewer 3 Report

The manuscript aimed to assess the prevalence and to quantify Salmonella and indicator microorganisms in pork processing stages to develop a baseline and stablish statistical process control parameters for safety management in pork processing plants. The bio-mapping study brings value on the identification of safety risks stages of the pork processing line and can support food safety management decision based on the statistical data on the prevalence and count of pathogens and indicator microorganisms.

The experiment seems to be well conducted and the results are properly discussed. Some points of clarifications and suggestions are indicated below.

 Line 100: please indicate the full description of abbreviations AC, EB, EC the first time they appear in the text,

Tables and Figures: Please describe the meaning of the abbreviations in the footnotes (Ex: PAA, AMR, etc.)
